# High Tumoral STMN1 Expression Is Associated with Malignant Potential and Poor Prognosis in Patients with Neuroblastoma

**DOI:** 10.3390/cancers15184482

**Published:** 2023-09-08

**Authors:** Kenjiro Ogushi, Takehiko Yokobori, Sumihito Nobusawa, Takahiro Shirakura, Junko Hirato, Bilguun Erkhem-Ochir, Haruka Okami, Gendensuren Dorjkhorloo, Akira Nishi, Makoto Suzuki, Sayaka Otake, Hiroshi Saeki, Ken Shirabe

**Affiliations:** 1Department of General Surgical Science, Graduate School of Medicine, Gunma University, Maebashi 371-8511, Japan; m1920018@gunma-u.ac.jp (K.O.); m2220010@gunma-u.ac.jp (H.O.); m2220603@gunma-u.ac.jp (G.D.); suzuki-m@gunma-u.ac.jp (M.S.); sayaka_oh@gunma-u.ac.jp (S.O.); h-saeki@gunma-u.ac.jp (H.S.); kshirabe@gunma-u.ac.jp (K.S.); 2Division of Integrated Oncology Research, Initiative for Advanced Research (GIAR), Gunma University, Maebashi 371-8511, Japan; bilguun.e@gunma-u.ac.jp; 3Department of Human Pathology, Gunma University Graduate School of Medicine, Maebashi 371-8511, Japan; nobusawa0319@gunma-u.ac.jp (S.N.); m1920028@gunma-u.ac.jp (T.S.); 4Department of Pathology, Public Tomioka General Hospital, Tomioka 370-2393, Japan; hirato-junko@gunma-u.ac.jp; 5Department of Surgery, Gunma Children’s Medical Center, Shibukawa 377-8577, Japan; anishi@gcmc.pref.gunma.jp; 6Department of Surgery, Iwate Medical University School of Medicine, Morioka 028-3695, Japan

**Keywords:** neuroblastoma, oncoprotein 18, prognostic markers, STMN1

## Abstract

**Simple Summary:**

This study found that high levels of STMN1 expression in neuroblastoma indicate malignant potential, proliferation potency, and poor prognosis. STMN1 knockdown inhibited neuroblastoma cell growth regardless of MYCN overexpression. The study suggests that assessing STMN1 expression could be a powerful indicator of prognosis and a promising therapeutic candidate against refractory neuroblastoma.

**Abstract:**

Background. Stathmin 1 (STMN1), a marker for immature neurons and tumors, controls microtubule dynamics by destabilizing tubulin. It plays an essential role in cancer progression and indicates poor prognosis in several cancers. This potential protein has not been clarified in clinical patients with neuroblastoma. Therefore, this study aimed to assess the clinical significance and STMN1 function in neuroblastoma with and without MYCN amplification. Methods. Using immunohistochemical staining, STMN1 expression was examined in 81 neuroblastoma samples. Functional analysis revealed the association among STMN1 suppression, cellular viability, and endogenous or exogenous MYCN expression in neuroblastoma cell lines. Result. High levels of STMN1 expression were associated with malignant potential, proliferation potency, and poor prognosis in neuroblastoma. STMN1 expression was an independent prognostic factor in patients with neuroblastoma. Furthermore, STMN1 knockdown inhibited neuroblastoma cell growth regardless of endogenous and exogenous MYCN overexpression. Conclusion. Our data suggest that assessing STMN1 expression in neuroblastoma could be a powerful indicator of prognosis and that STMN1 might be a promising therapeutic candidate against refractory neuroblastoma with and without MYCN amplification.

## 1. Introduction

Neuroblastoma (NB) is a common pediatric extracranial solid tumor. In clinical practice, the diagnosis of high-risk NB requires the presence of several factors such as MYCN (proto-oncogene BHLH transcription factor) amplification and copy number variation [1]. However, the prognosis of high-risk NB patients remains poor despite a multidisciplinary approach to treatment [2,3]. On the contrary, certain cases of NB show treatment resistance, although the current system of diagnosis does not identify such cases as high-risk [4,5]. Therefore, research and development of new biomarkers and therapeutic targets are necessary to diagnose tumor aggressiveness correctly and to improve prognoses in patients with refractory NB.

Stathmin 1 (STMN1), also known as oncoprotein 18, is a cytosolic phosphoprotein that controls microtubule dynamics by preventing tubulin polymerization and facilitating microtubule destabilization [6]. STMN1 is a neural differentiation marker [7,8]. It is overexpressed in several human malignancies [9]. High STMN1 expression in tumor tissues correlates with tumor aggressiveness, poor prognosis, and therapeutic resistance in several cancers [10,11,12,13,14,15,16,17]. These findings indicate that STMN1 may be a potential biomarker and therapeutic target in different types of primary tumors, including NB.

STMN1 could be upregulated in some chemo-resistant NB cell lines and could also be related to NB cell migration ability [18,19]. Its suppression by RNA interference (RNAi) caused an inhibitory effect on metastatic potential in a NB xenograft mouse model [20]. Moreover, it has been reported that STMN1 phosphorylation was associated with MYCN amplification in NB [21]. However, the importance of this potential protein as a tumor promotion factor, prognostic biomarker, and therapeutic candidate has not been elucidated in clinical NB with and without MYCN amplification.

This study aimed to evaluate the clinical significance of STMN1 expression in NB by examining the association of tumoral STMN1 expression levels with various clinicopathological parameters, including MYCN status and survival. Functional analysis of STMN1 was also performed in NB cell lines, with and without MYCN overexpression, to determine whether STMN1 targeting is promising for patients with refractory NB.

## 2. Materials and Methods

### 2.1. Patients and Samples

Surgical specimens were obtained from 102 NB-suspected patients who underwent surgical biopsy or curative surgery at Gunma University Hospital and Gunma Children’s Medical Center between 1991 and 2020. Eleven patients had received preoperative chemotherapy; hence, they were excluded from further analysis. Among the patients without preoperative chemotherapy, 81 were pathologically diagnosed with NB, 9 with ganglioneuroblastoma, and 1 with ganglioneuroma (Appendix A). Among the 81 NB tumor samples, 69 cases were non-MYCN-amplified, and 8 cases were MYCN-amplified. The MYCN status in four cases was not evaluated. In addition, the 81 patients with NB had 18 postoperative complications, including the ileus, renal dysfunction, liver disorder, chylothorax, sepsis, hearing disorder, Horner’s syndrome, herpes zoster, pseudomembranous enteritis, and opsoclonus-myoclonus syndrome. This study was conducted in accordance with the tenets of the Declaration of Helsinki and was approved by the Institutional Review Board for Clinical Research of Gunma University Hospital (Maebashi, Gunma, Japan; approval number: HS2020-069). Patient consent was obtained via the opt-out method.

The SK-N-AS (non-MYCN-amplified) cell line was purchased from the American Type Culture Collection. The NB69 (non-MYCN-amplified) cell line and the LAN-5 (MYCN-amplified) cell line were purchased from RIKEN Cell Bank. These are all human NB cell lines.

SK-N-AS cells were cultured in Dulbecco’s modified eagle medium (Wako, Osaka, Japan) supplemented with 10% fetal bovine serum and 1% penicillin–streptomycin, and NB69 and LAN-5 were used as the Roswell Park Memorial Institute (RPMI-1640) medium (Wako, Osaka, Japan); 10% fetal bovine serum and 1% penicillin–streptomycin were added. Cultured cells were incubated at 37 °C in a humidified atmosphere containing 5% carbon dioxide.

### 2.2. Immunohistochemistry

Paraffin-embedded specimens, including 81 NB, 9 ganglioneuroblastoma, and 1 ganglioneuroma, were sliced into 3.5 µm thick sections and mounted on glass slides. Then, these sections were incubated at 60 °C for 60 min, followed by deparaffinization with xylene and rehydration. Freshly prepared 0.3% hydrogen peroxide in 100% methanol was used to incubate the specimens for 30 min at room temperature to block endogenous peroxidase activity. After rehydration through a graded series of ethanol treatments, antigens were retrieved using an Immunosaver (Nishin EM, Tokyo, Japan) at 98 °C–100 °C for 45 min. The sections were passively cooled to room temperature and then incubated in a protein block serum-free reagent (DAKO, CA, USA) for 30 min. Then, tumor sections were incubated with a mouse monoclonal STMN1 antibody (1:200; Santa Cruz Biotechnology, CA, USA, sc-48362) in a Dako REAL antibody diluent at 4 °C for 24 h. According to the manufacturer’s instructions, Histofine Simple Stain MAX-PO (Multi) Kit (Nichirei, Tokyo, Japan) was used to visualized primary antibody staining. The sections were stained with 3,3-diaminobenzidine tetrahydrochloride (0.02% in 50 mM ammonium acetate–citrate acid buffer containing 0.005% hydrogen peroxide), lightly counterstained with hematoxylin, and mounted. Negative controls were not treated with the primary antibody. Immunohistochemical slides were evaluated by two experienced pathologists blinded to the clinical data. The staining score for each sample was the average of two pathologists’ evaluations.

### 2.3. Evaluation of Immunostaining

Immunoreactivity of STMN1 and Ki67 was assessed as follows: The positive cell number of STMN1 and Ki67 was calculated as the percentage of cytoplasmic and nuclear-stained cells for each section based on 500 NB cells and was counted at the site with the maximum amount of positive stain in the slide. The patients with NB were classified as high or low groups based on the receiver operating characteristic (ROC)-derived cut-off value of STMN1 and the number of Ki67-positive cells regarding poor prognosis. Based on the number of the ROC-derived cut-off value of STMN1 and Ki67-positive cells, the tumors were classified as high or low-STMN1-expression groups. Accordingly, a disease prognosis was made. A STMN1-positive cell number of ≥255 was defined as the high-STMN1-expression group, and <255 was defined as the low-STMN1-expression group. In addition, A Ki67-positive cell number of ≥88 was defined as the high-Ki67-expression group, and <88 was defined as the low-Ki67-expression group.

### 2.4. siRNA Transfection

STMN1-specific siRNA oligos (STMN1 siRNA1: GAAACGAGAGCACGAGAAAtt, STMN1 siRNA2: CGAGACUGAAGCUGACUAAtt) and non-targeting control siRNA oligos (control siRNA) were purchased from Theoria Science (Tokyo, Japan). First, target cell lines were subjected to siRNA using in vitro electroporation. Cells were suspended in serum-free Opti-MEM I (Life Technologies, Tokyo, Japan) at a density of 1 × 10^7^ cells/mL. Then, 1.5 µM siRNA was added to the cell suspension. A total of 200 µM of the cell suspension was transferred to a 2 mm gap cuvette electrode and electroporated with an electroporator (CUY21EDIT II; BEX Co., Tokyo, Japan).

### 2.5. Establishment of MYCN-Overexpressing NB Cell Lines

Non-MYCN-amplified NB cell lines, including SK-N-AS and NB69, were transduced to overexpress exogenous MYCN. Mock lentiviral vectors and the MYCN overexpression lentiviral vector pLV[Exp]-EGFP:-T2A:Puro-CMV>hMYCN were constructed and packaged by VectorBuilder (Chicago, IL, USA). SK-N-AS and NB69 cells were treated with viral particles in accordance with the manufacturer’s protocol and selected using 4.0 μg/mL of puromycin. Detailed information about the vector can be obtained from https://www.vectorbuilder.jp/ accessed on 14 June 2021, using the ID VB210615-1342kuf.

### 2.6. Western Blot Analysis

Western blot analysis was performed to confirm the expression of STMN1 (anti-STMN1 mouse monoclonal antibody, 1:1000; Santa Cruz Biotechnology, CA, USA sc-48362) and MYCN (anti-MYCN rabbit polyclonal antibody, 1:1000; Proteintech, 10159-2-AP). In addition, 10 μg of the total protein was loaded per lane, and β-actin expression (anti-β-actin mouse monoclonal antibody, 1:1000; Sigma-Aldrich, St. Louis, MO, USA) was used as a protein-loading control. The blots were detected using the ECL Western Blot Analysis Detection System and an Image Quant LAS 4000 machine (GE Healthcare Life Sciences, Westborough, MA, USA).

### 2.7. Cell Proliferation Assay

The proliferation of NB cell lines, with and without MYCN overexpression, treated with control or STMN1 siRNA was assessed. Cells were seeded in 96-well plates (SK-N-AS: 5000 cells, NB69: 5000 cells, LAN-5: 20,000 cells per well in 100 μL of medium containing 10% FBS). Cell proliferation was measured at 0, 24, 48, and 72 h after seeding with a Cell Counting Kit-8 (Dojindo Laboratories, Tokyo, Japan). A total of 10 µL of the solution Cell Counting Kit-8 was added to each well and incubated for 2 h at 37 °C. The absorbance of each well was measured with a microplate reader (Thermo, Waltham, MA, USA) at 450 nm.

### 2.8. Statistical Analysis

Group differences were evaluated using the Wilcoxon or the Chi-square test. ROC curve analyses were used to define the suitable cut-off value of STMN1 and Ki67 immunohistochemical evaluation to predict poor prognosis (overall death). Kaplan–Meier curves were generated for overall survival, and statistical significance was determined using the log-rank test. Univariate and multivariate survival analyses were conducted using the Cox proportional hazard regression model. In vitro data in three or four groups were analyzed using ANOVA. When the results of the ANOVA were significant, Dunnett’s multiple comparison tests were used to assess differences among groups. A *p*-value of <0.05 was considered statistical significance. All statistical analyses were performed with JMP Pro 14 software (SAS Institute, Cary, NC, USA).

## 3. Results

### 3.1. Immunohistochemical Staining of STMN1 in Clinical NB Tissues

In NB tumor sections, STMN1 was expressed in the cytoplasm of neuroblasts and a neuropil-like background (Figure 1) in a relatively uniform pattern. Of the 91 NB-suspected specimens, the ratio of positive STMN1 expression was higher in NB (23.5%, 19/81 cases) than in ganglioneuroma (0%, 0/9 cases), ganglioneuroblastoma (0%, 0/1 case), and surrounding non-tumoral tissues (Figure 1 and Appendix A). Of the 81 NB-diagnosed specimens, 62 (77%) and 19 (23%) were categorized as the low-STMN1-expression (Figure 1C) and high-STMN1-expression groups (Figure 1D), respectively (Table 1).

### 3.2. Association of STMN1 Expression with the Clinicopathological Features of Clinical Neuroblastoma Patients

The relationship between STMN1 and patient clinicopathological characteristics is shown in Table 1. High STMN1 expression was significantly associated with age; radical resection; proliferation marker Ki-67 expression; and the progression of INRGSS, INPC, and INRG risk groups (Table 1, left part). In patients with non-MYCN-amplified NB (*n* = 69), high STMN1 expression showed similar expression significance (Table 1, middle part). Regarding MYCN-amplified cases (*n* = 8), the number of cases in our cohort was small; hence, the statistical significance of STMN1 expression could not be demonstrated (Table 1, right part).

### 3.3. Prognostic Significance of STMN1 Expression in NB Patients with and without MYCN Amplification

Kaplan–Meier analysis of data from 81 patients with NB revealed significantly lower overall survival in the high-STMN1-expression group than in the low-expression group (*p* < 0.001, Figure 2A, left panel). Moreover, among the 69 patients with non-MYCN-amplified NB, the high-STMN1-expression group had poor prognosis compared with the low-expression group (*p* = 0.0049, Figure 2A, middle panel). In contrast to overall NB cases and non-MYCN-amplified ones, fewer MYCN-amplified NB cases (*n* = 8; *p* = 0.765, Figure 2A, right panel) hindered the drawing of any statistically significant conclusion about the prognostic value of STMN1 expression.

In confirming the prognostic significance of STMN1 expression in a larger cohort, the R2 Genomics Analysis and Visualization Platform with the prognostic data of 782 patients with NB was used (R2 internal identifier: ps_avgpres_dgc2102a786_dgc2102). The results show that the high-STMN1-expression group had poorer prognosis than the low-expression group in not only all cases (*n* = 782, *p* = 0.019) and non-MYCN-amplified cases (*n* = 629) (*p* < 0.001), but also MYCN-amplified cases (*n* = 153, *p* = 0.044). By using a large NB cohort from the R2 database, high STMN1 expression in NB, with or without MYCN amplification, provides statistically significant poor prognosis.

Our univariate analysis of 81 NB cases identified a high level of STMN1 expression as a significant prognostic factor associated with poor survival (hazard ratio = 6.439, 95% CI = 1.882–22.03, *p* = 0.003). Moreover, multivariate analysis identified high STMN1 expression as an independent risk factor for poor overall survival (hazard ratio = 16.95, 95% CI = 1.567–183.4, *p* = 0.019; Table 2).

### 3.4. STMN1 Suppression Inhibited Cellular Viability in NB Cell Lines with and without MYCN Amplification

STMN1 suppression by siRNA in non-MYCN-amplified SK-N-AS and NB69 cells was confirmed by Western blotting. The expression level of STMN1 protein was clearly suppressed in STMN1 siRNA groups compared with the control siRNA group (Figure 3A,B, left panel). Furthermore, significant reductions in cellular viability were observed in STMN1-siRNA-treated (suppressed) cells compared with control-siRNA-treated cells (Figure 3A right panel). In confirming the growth inhibition effect by STMN1 suppression against MYCN-amplified cells, cellular viability assay was performed using MYCN-amplified NB cell line LAN-5 cells. The viability in STMN1-siRNA-treated LAN5 cells was significantly suppressed compared with that in control-siRNA-treated cells (Figure 3B, right panel). In addition, STMN1 siRNA treatment against LAN-5 cells suppressed not only the target STMN1 protein but also the endogenous MYCN protein (Figure 3B, left panel), which plays a key role in NB aggressiveness.

In validating the interesting relationship between STMN1 suppression and endogenous MYCN downregulation, exogenous MYCN (MYCN OE) was overexpressed in non-MYCN-amplified cell lines SK-N-AS and NB69. Consistent with the MYCN reduction in STMN1-suppressed LAN-5 cells, the expression level of exogenous MYCN was lower in MYCN OE STMN1 siRNA-treated cells compared with MYCN OE control siRNA-treated cells (Figure 4A).

These experiments were performed in triplicate.

Moreover, the STMN1 siRNA treatment of MYCN OE cells could counteract MYCN OE-induced proliferation to a certain extent (Figure 4B).

## 4. Discussion

In this study, high levels of STMN1 expression were associated with tumor aggressiveness and poor prognosis in NB, and STMN1 expression was an independent prognostic factor in patients with NB. These results are consistent with those of previous studies of several other types of cancer [10]. Furthermore, STMN1 suppression can inhibit NB cell growth regardless of endogenous and exogenous MYCN overexpression.

The neuronal marker STMN1, or Oncoprotein18, is overexpressed in a wide variety of various tumor cells and nerve cancers. Many STMN1-inducible upstream factors, such as E2F [22], mutant p53 [23,24], PTEN loss [25], NF-κB [26], Sonic hedgehog signal [27], PPP1R14B [28], and diverse microRNAs [24,29,30,31,32], have been reported to promote aggressiveness in several cancers. STMN1 expression was upregulated in vincristine-resistant NB cells [18]. On the contrary, STMN1 expression levels are low in NB with Chromosome 1p36 (STMN1 locus) loss of heterozygosity (LOH), suggesting the existence of a regulatory mechanism of STMN1 expression in NB through genomic alterations [33]. In addition, the 1p36 LOH has been identified as an important locus alteration that is frequently observed in aggressive NB with MYCN amplification [34,35]. Previous studies show that aggressive NB has low STMN1 expression in 1p36 LOH cases with MYCN amplification, whereas our study shows high STMN1 expression as an independent poor prognostic factor. Therefore, STMN1 expression in MYCN-amplified and non-MYCN-amplified cases was compared using the R2 public database. As a result, STMN1 expression associated with poor prognosis was significantly lower in MYCN-amplified cases than in non-amplified cases (Appendix A). The data shown in Figure 2B indicate that the MYCN-amplified STMN1-expressing cases had a significantly poor prognosis. In using the neuronal marker STMN1 as a tumor marker for NB, the possibility of different regulatory mechanisms of STMN1 expression and appropriate cut-off values between low-risk NB and high-risk NB with 1p36 LOH and MYCN amplification should also be considered for risk classification.

Our in vitro data showed that STMN1 suppression downregulated endogenous and exogenous MYCN expression in NB cell lines. MYCN is an important factor that is strongly associated with malignant potential and poor prognosis in patients with NB. The development of methods and tools to regulate MYCN has been the focus of many researchers. To our knowledge, this report is the first to show MYCN expression regulation by STMN1. Our work shows that suppressing STMN1 lowers MYCN levels, indicating that STMN1 may be a novel upstream regulator of MYCN. Many investigators have examined therapeutic strategies to suppress STMN1, with some reporting that STMN1 function is activated by AKT, which can be an activation marker for the PI3K/AKT signaling pathway [25,36,37]. STMN1 silencing suppresses AKT activation in cancer cells [38,39]. In addition, highly aggressive neuroendocrine prostate cancer overexpresses MYCN, and AKT inhibition against such tumors could destabilize the MYCN protein [40]. We hypothesize that microtubule dynamics dysfunction is the main mechanism of the anti-tumor effect of STMN1 silencing. Moreover, the downregulation of the AKT/MYCN axis via STMN1-targeted therapeutic strategies can be a novel approach against refractory NB with MYCN amplification.

Database analysis indicated the significant prognostic value of STMN1 expression in NB with MYCN amplification (Figure 2B). Our in vitro data also showed that STMN1 suppression inhibited cellular viability in NB cells with endogenous and exogenous MYCN overexpression. These data strongly indicate the importance of STMN1 in refractory NB with and without MYCN amplification. We hypothesize that cell cycle dysregulation based on microtubule dynamics dysfunction is the main mechanism of the anti-tumor effect of STMN1 silencing. Moreover, the downregulation of the AKT/MYCN axis via STMN1-targeted therapeutic strategies can be a novel approach against refractory NB with MYCN amplification. Our study is the first report showing that assessing STMN1 protein expression by immunohistochemistry, frequently performed in clinical practice, may predict the clinical prognosis of patients with NB.

We think that immunostaining-based evaluation focusing only on neuroblasts can be used for STMN1-based prognosis for NB.

However, the sample size of MYCN-amplified cases in our cohort was insufficient to validate the potential of STMN1 protein assessment as a prognostic marker. Thus, further studies must be conducted to determine whether STMN1 protein expression assessment by immunohistological studies can also predict prognosis in NB with and without MYCN amplification.

This study had also some limitations. First, this study was retrospective in nature, and only operated patients without preoperative treatments were enrolled to know the significance of STMN1 baseline expression in NB. Second, the number of patients with resected samples was small in this study because NB is a rare type of malignant tumor, which may have introduced a bias in our results. Finally, functional analysis of STMN1 was not performed using RNAi in tumor-bearing mouse models because animal experiments have reported that STMN1 suppression by RNAi inhibited the metastatic potential in an orthotopic NB mouse model [19]. In developing effective STMN1 inhibition-based therapy for treating high-grade NB with MYCN amplification, the efficacy of STMN1 inhibitors and related tools should be investigated in cell and animal experiments.

## 5. Conclusions

In this study, high levels of STMN1 expression were shown to be a powerful marker of poor prognosis in patients with NB. STMN1 suppression could reduce the cellular viability and MYCN expression in NB cells. In developing new therapeutic tools, STMN1 might be a potential candidate for treating refractory NB with and without MYCN amplification.

## Figures and Tables

**Figure 1 cancers-15-04482-f001:**
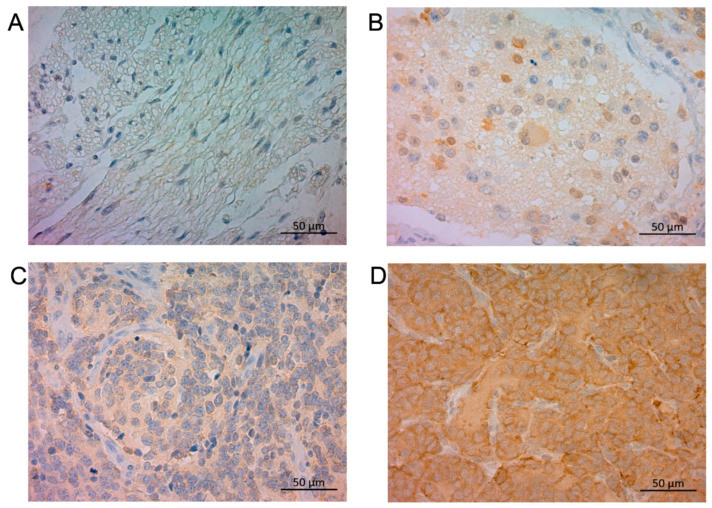
Immunohistochemical staining of STMN1 protein in ganglioneuroma, ganglioneuroblastoma, and neuroblastoma samples. Representative sections of (**A**) ganglioneuroma tissue without STMN1 expression, (**B**) ganglioneuroblastoma tissue with low STMN1 expression, (**C**) neuroblastoma tissue with low STMN1 expression, and (**D**) neuroblastoma tissue with high STMN1 expression. Scale bar = 50 μm.

**Figure 2 cancers-15-04482-f002:**
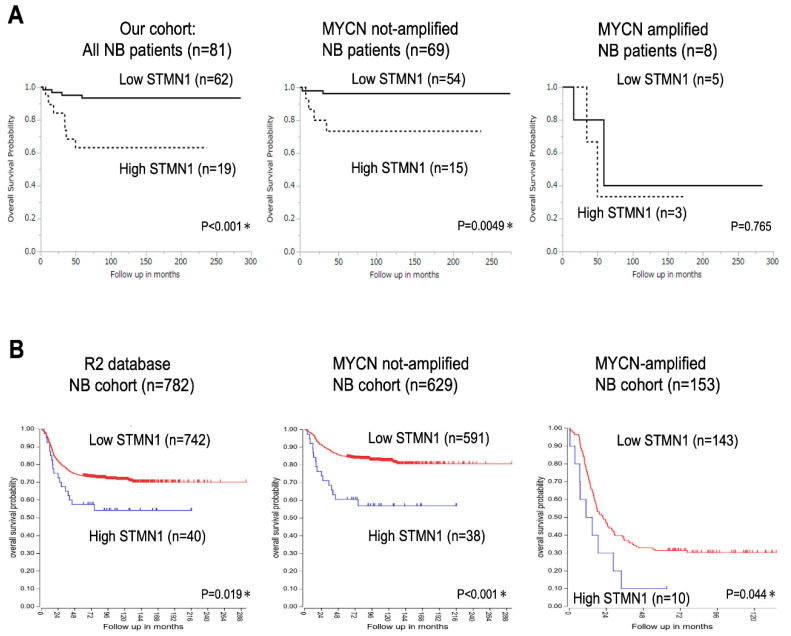
Kaplan–Meier survival curves of patients with neuroblastoma (NB) based on STMN1 expression. (**A**) Kaplan–Meier survival analysis based on SMNT1 expression, overall survival in our cohort of all NB patients (*n* = 81, *p* < 0.001, left panel), non-MYCN-amplified (*n* = 69, *p* = 0.0049, middle panel), and MYCN-amplified (*n* = 8, *p* = 0.765, right panel) NB cases. MYCN status in four cases was not evaluated. * *p* < 0.05. (**B**) Kaplan–Meier survival analysis based on SMNT1 expression in the R2 database of all NB (*n* = 782, *p* = 0.019, left panel), non-MYCN-amplified (*n* = 629, *p* < 0.001, middle panel), and MYCN-amplified (*n* = 153, *p* = 0.044, right panel) NB patients. We used an online R2 Genomics Analysis and Visualization Platform to verify the relevance of STMN1 mRNA expression to the overall survival of patients with NB: R2 internal identifier, ps_avgpres_dgc2102a786_dgc2102.

**Figure 3 cancers-15-04482-f003:**
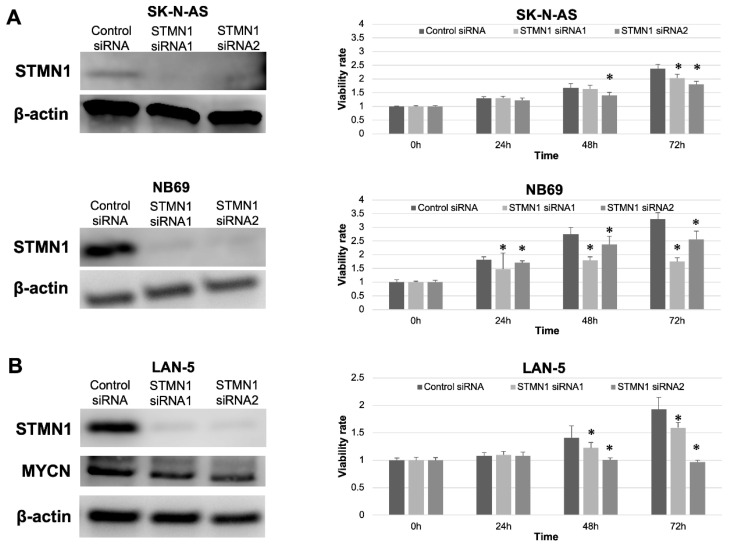
STMN1 knockdown inhibited cellular viability in non-MYCN-amplified and MYCN-amplified neuroblastoma cell lines. Cell lines were transfected with STMN1-specific siRNAs and control siRNA. β-actin expression was used as gel loading control. Cellular viability was verified using the Cell Counting Kit-8. (**A**) Left panel: Western blot showing a decrease in STMN1 expression levels in lysates from non-MYCN-amplified SK-N-AS and NB69 neuroblastoma cells. Right panel: cellular viability. (**B**) Left panel: Western blot showing a decrease in STMN1 and endogenous MYCN expression levels in lysates from MYCN-amplified LAN-5 neuroblastoma cells. Right panel: Cellular viability. The control siRNA group was defined as the control for Dunnett’s multiple comparison tests. * *p* < 0.05. All WB images are included in File S1.

**Figure 4 cancers-15-04482-f004:**
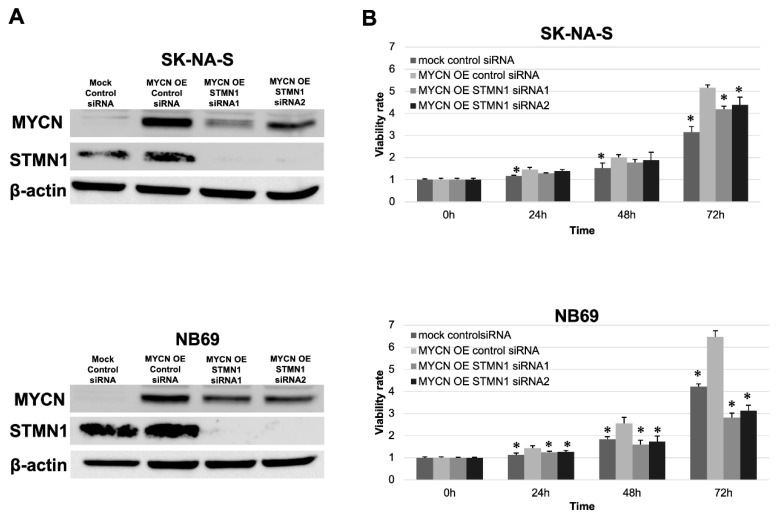
STMN1 suppression inhibited the MYCN expression and MYCN-induced proliferation in neuroblastoma cell lines overexpressing exogenous MYCN. Non-MYCN-amplified SK-N-AS and NB69 neuroblastoma cell lines were transduced for overexpressing exogenous MYCN (MYCN OE). (**A**) Western blot showing a decrease in STMN1 and exogenous MYCN expression levels in lysates from cells transduced with STMN1-specific siRNAs compared with the negative control siRNA transfectants. β-actin expression was used as gel loading control. These experiments were performed in triplicate. (**B**) MYCN-induced proliferation in cells with MYCN OE or MYCN OE + STMN1 siRNAs. Cell viability in control siRNA, MYCN OE, and MYCN OE + STMN1 siRNA groups was verified using the Cell Counting Kit-8. The MYCN OE control siRNA group was defined as the control for Dunnett’s multiple comparison tests. * *p* < 0.05. These experiments were performed in triplicate. All WB images are included in File S1.

**Table 1 cancers-15-04482-t001:** The relationship of clinicopathlogical factors and STMN1 expression in NB patients.

Factors	Total NB Cohort (*n* = 81)		*p* Value	MYCN Not-Amplified (*n* = 69)		*p* Value	MYCN Amplified (*n* = 8)		*p* Value
	Low	High		Low	High		Low	High	
	*n* = 62	*n* = 19		*n* = 54	*n* = 15		*n* = 5	*n* = 3	
Gender									
Male	27	10	0.6	23	8	0.561	3	1	1
Female	35	9		31	7		2	2	
Age, month									
<18 m	56	11	0.002 *	52	10	0.004 *	1	1	1
18 m≤	6	8		2	5		4	2	
Distant metastasis									
Absent	26	4	0.113	26	4	0.156	0	0	-
Present	36	15		28	11		5	3	
INRGSS									
L1, L2, MS	50	7	0.001 *	46	7	0.004 *	1	0	1
M	12	12		8	8		4	3	
Radical resection									
No	21	16	<0.001 *	16	12	<0.001 *	5	3	-
Yes	41	3		38	3		0	0	
INPC									
Favorable	52	7	<0.001 *	48	7	0.001 *	1	0	1
Unfavorable	10	12		6	8		4	3	
MYCN status									
Not amplified	54	15	0.38						
Amplified	5	3							
INRG risk group									
Not high	55	12	0.016 *	52	11	0.017 *	0	0	-
High	7	7		2	4		5	3	
Chemotheraphy									
No	23	3	0.098	23	3	0.139	0	0	-
Yes	39	16		31	12		5	3	
Ki67 expression									
Low	42	6	0.007 *	37	5	0.018 *	4	1	0.464
High	20	13		17	10		1	2	

INRGSS: International neuroblastoma risk group staging system. INPC: international neuroblastoma pathology classification. * Significant difference *p* < 0.0.5.

**Table 2 cancers-15-04482-t002:** Univariate and multivariate analyses of overall survival in 81 NB patients.

Factors		Univariate Analysis		Multivariate Analysis	
		Hazard Ratio [95% Confidence Interval]	*p* Value	Hazard Ratio [95% Confidence Interval]	*p* Value
Gender	Female	1	0.196	1	0.015 *
	Male	2.247 [0.657–7.678]		21.56 [1.814–256.2]	
Month at diagnosis	<18	1	<0.001 *	1	0.158
	≥18	17.92 [4.689–68.51]		0.074 [0.001–2.766]	
Distant metastasis	Absent	1	0.999	1	0.999
	Present	3.06 × 10^9^		4.63 × 10^9^	
INRGSS	L1, L2, MS	1	0.013 *	1	0.562
	M	29.26 [3.738–229]		0.372 [0.013–10.5]	
Radical resection	No	1	0.999	1	0.999
	Yes	1.53 × 10^−10^		6.06 × 10^−9^	
INPC	FH	1	0.006 *	1	0.46
	UH	14.51 [3.12–67.45]		0.31 [0.014–6.9]	
MYCN status	Not amplified	1	0.0029 *	1	0.711
	Amplified	6.937 [1.937–24.83]		0.714 [0.12–4.242]	
INRG risk group	Not high	1	<0.001 *	1	0.031 *
	High	18.19 [4.757–69.62]		199 [1.603–24726]	
Chemotherapy	No	1	0.999	1	1
	Yes	2.62 × 10^9^		0.477	
Complication	Absent	1	0.006 *	1	0.022 *
	Present	5.24 [1.595–17.21]		9.985 [1.383–72.04]	
Ki67 expression	Low	1	0.046 *	1	0.035 *
	High	3.491 [1.021–11.936]		9.402 [1.167–75.74]	
STMN1 expression	Low	1	0.003 *	1	0.019 *
	High	6.439 [1.882–22.03]		16.95 [1.567–183.4]	

* Significant difference *p* < 0.05.

## Data Availability

All data generated or analyzed in this study are included in this publication and its Appendix A data.

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
