# Peer review of "High Tumoral STMN1 Expression Is Associated with Malignant Potential and Poor Prognosis in Patients with Neuroblastoma"

_cancers, 2023, doi:10.3390/cancers15184482_

Round 1
Reviewer 1 Report
This manuscript addresses the potential prognostic significance of high STMN1 expression in neuroblastoma tumours and the influence of STMN1 over neuroblastoma cell proliferation in culture. Historically STMN1 is implicated as a pro-tumorigenic gene in a number of cancers and there is only a small amount of data supporting this in neuroblastoma models. The authors here wanted to pursue the role of STMN1 using both clinical data and also culture models of non-MYCN-amplified tumours.
This is an interesting piece of work and shows from a study of clinical samples that high STMN1 protein levels correlate with poor prognosis in the patient group, particularly in non-MYCN amplified cases. When survival data are analysed, again high STMN1 correlated with worse survival in the non-MYCN amplified and amplified groups. STMN1 appears to be an independent marker of disease outcome, suggesting that it could be useful in future as a potential prognostic tool. In cultured neuroblastoma cell lines, the authors demonstrate that loss of STMN1 after siRNA treatment would reduce cell proliferation in both MYCN-amplified and non-amplified cells. In addition they provide the first evidence that suppression of STMN1 expression results in concomitant lowering of MYCN protein levels. This also occurred in cells forced to express high MYCN levels.
Overall this study is well designed and executed and the data are well presented. There are nevertheless some areas that need clarification.
1. The data in figure 2 are clear. However, it will be important to explain exactly what datasets were included in the R2 study. The result is strange because STMN1 expression in several large neuroblastoma datasets in R2 clearly correlate in the opposite fashion (eg Kocak, SEQC; overall survival Kaplan-Meier). The authors should perhaps present more R2 datasets, and must discuss these discrepancies.
2. Figure 3. The data demonstrate reduced cell numbers after STMN1 suppression in three quite distinct cell lines. The data do not show if there is a loss of cell viability (ie apoptosis) or just a reduced proliferation rate. Could the authors expand on this? In the Discussion they mention reduced “viability”, but this may not be strictly accurate.
3. Figures 3 and 4. The relationship between STMN1 and MYCN is novel and warrants further study at the molecular level. The authors should present quantitation of gels to make it clear how MYCN protein is reduced. The number of repeats carried out should also be made clear. The authors could also examine the levels of pAKT, pERK and pAURKA for example in these lysates, so begin to dissect the molecular basis of this reduced MYCN.
4. The authors should note the experimental repeat numbers (biological not technical) for legends in Figures 3 and 4.
5. There are a couple of further references that may be of help to list :
a. N. Hailat, J. Strahler, R. Melhem, X.X. Zhu, G. Brodeur, R.C. Seeger, C. P. Reynolds, S. Hanash, N-myc gene amplification in neuroblastoma is associated with altered phosphorylation of a proliferation related polypeptide (Op18), Oncogene 5 (1990) 1615–1618.
b. S.T. Po'uha, M. Le Grand, M.B. Brandl, A.J. Gifford, G.J. Goodall, Y. Khew-Goodall, M. Kavallaris, Stathmin levels alter PTPN14 expression and impact neuroblastoma cell migration, Brit J Cancer 122 (2020) 434–444.
Author Response
Response to the reviewers’ comments
Reviewer: 1
Query 1.
The data in figure 2 are clear. However, it will be important to explain exactly what datasets were included in the R2 study. The result is strange because STMN1 expression in several large neuroblastoma datasets in R2 clearly correlate in the opposite fashion (eg Kocak, SEQC; overall survival Kaplan-Meier). The authors should perhaps present more R2 datasets, and must discuss these discrepancies.
Answer 1.
Thank you for your suggestion.
For the R2 data analysis, we selected the dataset with the largest number of registered neuroblastoma cases (Cangelosi). First, we found a cutoff (82.9951) where STMN1 is high in all cases and the prognosis was significantly worse. Then we used that cutoff to confirm that high STMN1 cases had worse prognosis in MYCN non-amplified cases as well. On the other hand, we tried to use the same cutoff for MYCN-amplified cases, but the same cutoff could not be used because the expression of STMN1 in MYCN-amplified was significantly lower than that in MYCN not-amplified, as shown in Supplementary Material 2. An alternative cutoff (58.3882) was used to create the Kaplan-Meier curve for cases with MYCN amplification.
In the default analysis of R2, there are some cohorts, as pointed out by the referee, that showed the opposite results as a better prognosis in the STMN1-high expressing group. Changing cut off did not result in a significantly worse prognosis with STMN1 high in these two cohorts.
We think that the R2 data shows the expression levels of STMN1 mRNA in the cytoplasm of neuroblasts and a neuropil-like background where STMN1 is, and that this is one reason for the differences in the strength of STMN1 based prognosis expression in other large scale neuroblastoma data sets in R2. We propose that STMN1 is a prognostically relevant factor in clinical specimens of neuroblastoma if the cytoplasm of neuroblastoma cells alone is considered for appropriate evaluation of STMN1 by immunostaining and by objective measurement by a pathologist with appropriate cutoff.
When considering STMN1 as a biomarker, evaluation of STMN1 mRNA levels is not reliable. We think that only immunostaining and evaluation, focusing only on neuroblasts is important.
Query 2.
Figure 3. The data demonstrate reduced cell numbers after STMN1 suppression in three quite distinct cell lines. The data do not show if there is a loss of cell viability (ie apoptosis) or just a reduced proliferation rate. Could the authors expand on this? In the Discussion they mention reduced “viability”, but this may not be strictly accurate.
Answer 2.
Thank you for your suggestion.
As the referee's comment mentions, our data did not show why the STMN1-suppressed NB cells had low cellular viability as measured by CCK8 assay. However, many researchers already reported the functional importance of STMN1 on cell cycle and proliferation. Therefore, to clarify this point, we revised the discussion part as follows:
"We hypothesize that cell cycle dysregulation based on the microtubule dynamics dysfunction is the main mechanism of the anti-tumor effect of STMN1 silencing. Moreover, the downregulation of the AKT/MYCN axis by STMN1-targeted therapeutic strategies can be a novel approach against refractory NB with MYCN amplification."
Query 3.
Figures 3 and 4. The relationship between STMN1 and MYCN is novel and warrants further study at the molecular level. The authors should present quantitation of gels to make it clear how MYCN protein is reduced. The number of repeats carried out should also be made clear. The authors could also examine the levels of pAKT, pERK and pAURKA for example in these lysates, so begin to dissect the molecular basis of this reduced MYCN.
Answer 1.
Thank you for your suggestion. To explain the mechanism of MYCN supexpression by STMN1 inhibition, we evaluated the alteration of several cancer-related proteins in STMN1-suppressed NB cells as the referee suggested: for instance, pAKT, mTOR, and FBXW7. These proteins have been reported to be MYCN down regulators. However, we could not get reproducible and clear results. Contrary to expectations, the upstream factors of MYCN analyzed were not regulated by STMN1. Therefore, we could not present data on the regulatory mechanism of MYCN expression by STMN1 in the results. Therefore, we only stated our hypothesis regarding the regulatory mechanism of MYCN by STMN1, in the discussion.
Query 4.
The authors should note the experimental repeat numbers (biological not technical) for legends in Figures 3 and 4.
Answer 4.
In response to this comment, we explained the repeat number in the legends of figure 3 and 4.
Query 5.
There are a couple of further references that may be of help to list :
- N. Hailat, J. Strahler, R. Melhem, X.X. Zhu, G. Brodeur, R.C. Seeger, C. P. Reynolds, S. Hanash, N-myc gene amplification in neuroblastoma is associated with altered phosphorylation of a proliferation related polypeptide (Op18), Oncogene 5 (1990) 1615–1618.
- S.T. Po'uha, M. Le Grand, M.B. Brandl, A.J. Gifford, G.J. Goodall, Y. Khew-Goodall, M. Kavallaris, Stathmin levels alter PTPN14 expression and impact neuroblastoma cell migration, Brit J Cancer 122 (2020) 434–444.
Answer 5.
Accordingly, we cited the references in the introduction.

Reviewer 2 Report
Well presented clinical of laboratory data for STMN1 that might lead to novel therapy for high-risk neuroblastoma.
Expansive clinical of laboratory data are well presented and very interested!
Author Response
Reviewer 2:
Well-presented clinical of laboratory data for STMN1 that might lead to novel therapy for high-risk neuroblastoma.
Reply:
Thank you for this positive comment for our research.